



# Indications of improved seasonal sea level forecasts for the United States Gulf and East Coasts using ocean-dynamic persistence

Xue Feng[1], Matthew J. Widlansky[1,2], Tong Lee[3], Ou Wang[3], Magdalena A. Balmaseda[4], Hao Zuo[4], Gregory Dusek[5], William Sweet[5], and Malte F. Stuecker[2,6]

[1]Cooperative Institute for Marine and Atmospheric Research, School of Ocean and Earth Science and Technology (SOEST), University of Hawai'i at Mānoa, Honolulu, HI, USA
[2]Department of Oceanography, SOEST, University of Hawai'i at Mānoa, Honolulu, HI, USA
[3]Jet Propulsion Laboratory, California Institute of Technology, Pasadena, CA, USA
[4]European Center for Medium-Range Weather Forecasts, Reading, UK
[5]National Ocean Service, NOAA, Silver Spring, MD, USA
[6]International Pacific Research Center, SOEST, University of Hawai'i at Mānoa, Honolulu, HI, USA

*Correspondence to*: Xue Feng (xfeng@hawaii.edu) and Matthew J. Widlansky (mwidlans@hawaii.edu)

**Abstract.** Forecasting seasonal sea levels along many coasts remains challenging, with generally lower skills than forecasts for the open oceans. We investigate the influence of ocean dynamics on forecasting monthly sea level anomalies for the United States Gulf and East Coasts using the Estimating Circulation and Climate of the Ocean (ECCO) system, which is initialized monthly from 1992 through 2017 and runs forward for 12 months under climatological atmospheric forcing. This approach, which we refer to as an ocean-dynamic persistence forecast, demonstrates improved skill compared to both observed-damped persistence and the ECMWF SEAS5 climate forecast system when evaluated against observations. At a lead of 4 months, dynamic persistence has the highest anomaly correlation coefficients at 22 out of 39 coastal locations (mostly south of Cape Hatteras). However, improvement in root-mean-square error is minimal, possibly due to reduced variability in ECCO associated with its climatology forcing and coarse resolution. This study suggests dynamic persistence offers the potential to improve sea level forecasts beyond the capabilities of damped persistence and a state-of-the-art climate model.

## 1 Introduction

Forecasting monthly changes in coastal sea levels is challenging, especially along the United States (U.S.) Gulf and East Coasts, where climate models have yet to demonstrate skill at seasonal lead times (Long et al., 2021; Long et al. in review). In the open ocean, using global climate models to forecast sea level variability has shown promising results (Widlansky et al., 2023; Balmaseda et al., 2024), particularly in the tropical Pacific where predictions of monthly sea level anomalies are routinely provided to island communities in the form of enhanced high-tide outlooks (Widlansky et al., 2017). Climate models such as the fifth-generation Seasonal Forecasting System (SEAS5) from the European Centre for Medium-Range Weather Forecasts (ECMWF) are also skillful for sea surface height (SSH) variability in much of the tropical and subtropical



Atlantic Ocean (Balmaseda et al., 2024), although no sea level prediction products utilizing this skill in the open-ocean Atlantic have yet to be developed for the coastal U.S.

The main reason why climate model forecasts of SSH in the Atlantic Ocean are not used to improve seasonal predictions of high-tide flooding, such as NOAA's monthly outlook, is because the state-of-the-art models are not skillful at Gulf and East Coast water level gauge locations (Long et al., 2021; Widlansky et al., 2023; Balmaseda et al., 2024). Consider Charleston, South Carolina, where the anomaly correlation coefficient (ACC) at seasonal lead times (e.g., the outlook 4 months in the future, which we call the lead-4 month forecast) is nearly zero if utilizing SSH from SEAS5 or any other climate model so

far assessed (Long et al. in review). Since climate models have yet to demonstrate skillful seasonal sea level forecasts for the Gulf and East Coasts, NOAA currently relies on a statistical understanding of the monthly anomalies (i.e., the autocorrelation damping timescale of water level observations; referred to as a damped-persistence forecast), combined with long-term trends and tide predictions, to provide information for their monthly high-tide outlooks (Dusek et al., 2022). Unfortunately, including damped persistence in the NOAA outlook minimally increases skill beyond a climatology-only

model for most locations on the Gulf and East Coasts because the statistical persistence of sea level anomalies dampens to near zero by the lead-4 month, especially between Key West, Florida, and Cape Hatteras, North Carolina where the autocorrelation decay is fastest (Dusek et al., 2022).

The apparent lack of skillful seasonal sea level forecasts for the Gulf and East Coasts contrasts with what we could expect

based on the relatively slowly evolving ocean, which provides sources of predictability on seasonal and longer timescales (e.g., Hasselmann, 1976; Deser et al., 2003). The persistence of oceanic conditions, due to the ocean's high inertia of thermal and momentum energy, or so-called ocean memory (e.g., Frankignoul and Hasselmann, 1977; Shi et al., 2022), allows sea level anomalies to be tracked in the open ocean for months (e.g., Chelton and Schlax, 1996). Slow ocean dynamics, such as westward-propagating Rossby Waves, sometimes carry sea level anomalies across the Atlantic, which project onto the

coastal sea level variability, especially south of Cape Hatteras (Minobe et al., 2017; Calafat et al., 2018; Dangendorf et al., 2023; Wang et al., 2024). In addition, advective processes in the ocean as well as coastal-trapped waves transport density perturbations from high latitudes southward along the East Coast, which impact sea levels along their path (Frederikse et al., 2017; Wang et al., 2022; Zhu et al., 2024). Considering these well-established physical processes and the availability of reliable initial conditions in this region from ocean reanalyses (Feng et al., 2024), it seems reasonable to expect the

possibility of achieving more skillful seasonal sea level forecasts, especially for the Southeast Coast. However, certain aspects of the coastal environment, such as the highly variable nature of nearshore winds (Lee et al., 2023), are likely to continue complicating efforts to skillfully forecast the sea level variability, as these winds can disrupt otherwise predictable oceanic conditions and influence local sea level responses.



One opportunity for improving sea level forecasts is to isolate oceanic information from atmospheric variability; the latter of which is much less predictable at leads beyond a couple of weeks (e.g., Hasselmann, 1976; Newman et al., 2003), then let the ocean evolve according to its internal dynamics. Using an adjoint sensitivities analysis derived from the Estimating Circulation and Climate of the Ocean (ECCO) model, Frederikse et al. (2022) evaluated a hybrid dynamical forecasting method consisting of atmospheric forcing from either a climate model forecast (CCSM4) participating in the North

American Multi-Model Ensemble or the mean annual cycle (i.e., climatology). They referred to the latter forcing approach as "ocean-dynamic persistence" and applied it to forecasting sea levels at one pilot location on the Southeast Coast (Charleston). At the lead-4 month, their study showed that both sets of retrospective forecasts had higher ACC values (and lower root-mean-square error; RMSE) than the damped-persistence forecast at the Charleston water level gauge, with the dynamic persistence approach performing best (see their Figure 6).


Given the promising result at Charleston in Frederikse et al. (2022), we aim to test the opportunity for expanding improvement to other parts of the Gulf and East Coasts using the dynamic-persistence approach. Our investigation utilizes a set of retrospective forecasts produced with an initialized version of ECCO that runs forward for 12 months under climatological atmospheric conditions, allowing us to evaluate the potential of ocean-dynamic persistence in forecasting sea

level anomalies along the coast at the model's resolution (details about the model and forecasting method are described in Section 2). We will compare the performance of the ECCO dynamic-persistence model with that of the observed-damped persistence of sea levels as well as the SSH forecast from the SEAS5 climate model. Through this assessment, we seek to address current limitations in skillfully forecasting coastal sea levels by exploring the potential of ocean-dynamic persistence.

**2 Data and methods**

This study utilizes sea level observations from NOAA's National Water Level Observation Network as well as various satellite altimetry missions to assess the performance of different seasonal forecasting methods. For water level gauges, we obtained hourly sea level data from NOAA's Center for Operational Oceanographic Products and Services (CO-OPS) for 39 stations along the Gulf and East Coasts, consistent with the locations used in Feng et al. (2024) to assess coastal sea level

variability in ocean reanalyses. Since the present study focuses on the dynamic sea level forecast, the local inverse barometer (IB) effect is removed from the observations using sea level pressures from the ERA5 reanalysis (Hersbach et al., 2023). For altimetry, we obtained gridded absolute dynamic topography level 4 data from the Copernicus Marine and Environment Monitoring Service (CMEMS), which has a spatial resolution of 0.25°×0.25° and daily temporal resolution. A dynamic atmospheric correction has been applied to the CMEMS product so that there is no IB effect. We performed monthly

averaging of both observations (i.e., the hourly water levels and daily altimetry). These monthly sea level observations cover the overlapping period of the retrospective forecasts assessed here, which begin in 1993.



Observations from water level gauges and altimetry are used in the damped-persistence forecasts assessed here. In the damped-persistence model, each forecast is generated using the previous month's observations to predict the target month's sea level anomalies. The damping timescale is determined from the observed rate of how sea level anomalies decay to zero. We refer to the prediction for the month after the observation as the lead-1 month forecast. Note that the dynamical models considered here are initialized on the first day of each month, and we refer to the mean output during the first month of simulation as the lead-1 month forecast.

The dynamic-persistence forecasts are generated by NASA's Jet Propulsion Laboratory (JPL) using the ECCO system. The initial conditions of the forecast are derived from the observation-constrained ocean state estimate from ECCO Version 4 Release 4 (Forget et al., 2015), which is based on the global MITgcm with a 1° nominal resolution. The ocean-state estimate is obtained by optimizing a set of control variables (surface forcing, mixing parameters, and initial state) in an iterative process such that the forward model solution is consistent with a diverse set of satellite and in-situ ocean observations within the uncertainty estimates of the observations (Forget et al., 2015). The atmospheric state variables from the ECCO optimization are also used to force the dynamic-persistence model. For each month from January 1992 to December 2017, the model is integrated forward in time for 12 months with monthly climatological forcing to generate a set of 12-month forecasts. Because these forward simulations use climatological seasonal forcing, any interannual variations or monthly anomalies in the retrospective forecasts are due to the evolution of the initial ocean state. Note that the dynamic-persistence forecast is not an ensemble forecast, meaning that there is a single forecast for each target month at a specific lead time. We assessed the monthly dynamic sea level output, which does not include the IB effect.

The operational seasonal forecast system (SEAS5) at ECMWF is a fully coupled global climate model utilizing the NEMO ocean component with a nominal horizontal resolution of 0.25° (Johnson et al., 2019). This model was initialized monthly from January 1993 to December 2023 using the ECMWF's Ocean ReAnalysis System 5 (ORAS5; Zuo et al., 2019). For forecasts initialized in February, May, August, and November, SEAS5 provides a 13-month outlook with 15 ensemble members, while forecasts from other months have a 7-month outlook with 25 ensemble members. We evaluate the ensemble-mean forecast for the period overlapping the ECCO dynamic-persistence model (i.e., 1993–2017), although the SEAS5 assessment is focused on when forecasts are available from all 12 start months (i.e., lead times out to 7 months). SEAS5 excludes the IB effect.

We evaluated the seasonal forecast skills of the damped-persistence (observed), dynamic-persistence (ECCO), and climate-forecast (SEAS5) models by comparing their monthly sea level anomalies, nearest to the locations of water level gauges, against such observations as well as altimetry interpolated to the model grids. To correct for biases caused by model drift (see Widlansky et al., 2017; Long et al., 2021), the sea level anomaly forecasts from ECCO and SEAS5 were calculated by





removing the lead-time dependent monthly climatology of the retrospective epoch common to the forecast models (i.e., 1993–2017). Linear trends for that period were also removed from each forecast, respectively, which follows established methods for avoiding the influence of long-term changes on seasonal forecast skills (e.g., Widlansky et al., 2017; Long et al., 2021; Balmaseda et al., 2024; Long et al. in review). Observed sea level anomalies were calculated relative to climatology

based on the 1993–2017 period, with the trend removed accordingly. Metrics of assessment include standard deviation (SD), ACC, and RMSE, which are presented for each model as well as relative to the performance of the damped-persistence model. For the latter comparison, we tested the significance of ACC differences at each location using Fisher's Z transformation method. We calculated the z-statistic based on the difference between the two transformed z-scores, and a two-tailed test at the 0.05 significance level was applied against the null hypothesis that both models have similar skill. The

sample size was set to 294, which is the number of target months available for each forecast lead time.

## 3 Results

Seasonal forecasting skill according to the ACC metric is shown in Figure 1 across the three models: damped persistence (observed), dynamic persistence (ECCO), and climate forecast (SEAS5). The sea level anomalies at lead-1 and lead-4 months were evaluated against observations from water level gauges and satellite altimetry. Results for locations of The

Battery (New York; NY), Charleston (South Carolina; SC), Virginia Key (Florida; FL), and Grand Isle (Louisiana; LA) are shown to provide examples of the sea level forecast verification for the respective parts of the coastal U.S. (i.e., in the Northeast, Southeast, southeastern Florida, and the Gulf of Mexico).







**Figure 1.** Retrospective forecast skill (ACC) at lead-1 and 4 months for the damped-persistence (observed; a and b), dynamic-persistence (ECCO; c and d), and climate-forecast (SEAS5; e and f) models. Forecasts of monthly sea level anomalies are compared to observations by altimetry (shading) and water level gauges (colored circles on land) at four example locations (indicated by unfilled circles on the coastline). The domain-averaged ACC is shown in the upper left of each panel.



At the lead-1 month, damped persistence (using altimetry) demonstrates fairly high ACC throughout most open-ocean areas of the northwestern Atlantic and Gulf of Mexico (i.e., ACC values exceed 0.5 almost everywhere); however, ACC values are
lower along the coast (Figure 1a). For example, the ACC of damped persistence (using water level gauges) is 0.32 at The Battery, 0.40 at Charleston, 0.39 at Virginia Key, and 0.44 at Grand Isle. There are also some offshore areas of relatively lower ACC values (and higher RMSE; Figure 2), where the monthly sea level variability is much greater than at the coast (see Figure 1 in Long et al., 2021).

Dynamic persistence exhibits similar ACC values compared to damped persistence of altimetry and water level gauges at lead-1 month for the coastal examples; however, skill is somewhat lower in most of the open ocean (Figure 1c). The Loop Current region of the interior Gulf of Mexico as well as where the Gulf Stream extends away from the coast (i.e., at about 35 °N in the Atlantic Ocean) are notable examples of where dynamic persistence ACC values are much lower than damped persistence, at the lead-1 month. The RMSE metric mostly mirrors this result (i.e., higher errors in these energetic offshore
areas; Figure 2). In most other areas of the open ocean, ACC values are higher and RMSE is lower for all models relative to within the Loop Current/Gulf Stream system, although the overall skill at the lead-1 month for dynamic persistence is clearly worse than damped persistence nearly everywhere (considering the domain-average values listed in Figure 1 and 2).

The climate-forecast model has similar ACC values at the lead-1 month (Figure 1e), overall, to dynamic persistence (Figure
1c). Retrospective forecasts from both models exhibit relatively low ACC values in the interior Gulf of Mexico and the Gulf Stream extension region, especially compared to damped persistence (Figure 1a). The climate forecast performs particularly well for a broad area of the subtropical Atlantic Ocean, where its ACC values equal or beat damped persistence. The RMSE metric again mirrors the ACC result for the climate forecast (Figure 2). For the coastal locations, at the lead-1 month, ACC and RMSE values are similar among the three retrospective forecasts.





**Figure 2. Same as Figure 1, but for RMSE.**



By the lead-4 month, each model shows less forecast skill according to the ACC metric, both along the coast and in most offshore areas (Figure 1b, d, f). The decline in skill compared to the lead-1 month forecasts is worst for the damped-persistence model. In particular, the domain-average ACC for damped persistence has declined from 0.52 at the lead-1 month to 0.16 at the lead-4 month. Along the coast, there is no evidence of damped persistence having much skill at the lead-4 month, as the ACC is only 0.22 at The Battery, 0.14 at both Charleston and Virginia Key, and 0.16 at Grand Isle. In the offshore regions, ACC values are also generally less than 0.2, except for a region of noticeably higher values around Cuba and The Bahamas. RMSE is also low in these areas (Figure 2), which experience small sea level variability (Long et al. 2021).

Dynamic persistence is the best-performing model at the lead-4 month, according to the ACC metric, particularly along the coast (Figure 1d). Consider the forecasts at Charleston where the lead-4 month ACC value for dynamic persistence is 0.36, compared to only 0.14 or 0.08 for damped persistence or the climate forecast, respectively. However, RMSE assessments at the lead-4 month show only modest improvements for dynamic persistence compared to the other models (e.g., at Charleston, the RMSE is 6.3 cm for dynamic persistence, which is not much better than the 6.6 cm for damped persistence and 7.3 cm for SEAS5; Figure 2). Offshore, the lead-4-month RMSE values are likewise similar among the models (Figure 2). Notably, the climate forecast has lower ACC values along the coast at this lead (Figure 1f), compared to dynamic persistence, although the offshore performance of the former equals or beats the other models.

Using damped persistence as a benchmark of seasonal forecasting skill (or lack thereof), Figures 3 and 4 show lead-1 and lead-4 month ACC and RMSE differences, respectively, for the dynamic-persistence and climate-forecast models. At the lead-1 month, dynamic persistence performs worse than damped persistence along the Northeast Coast, while no significant differences are found along the Southeast and Gulf Coasts (Figure 3a, b). However, by the lead-4 month, the dynamic-persistence model has better forecast skill, with significantly higher ACC compared to damped persistence for most locations along the Southeast Coast (Figure 4a). The region with higher ACC for the dynamic-persistence model extends to the northeastern Gulf Coast, though the difference there is not statistically significant. However, the spatial pattern of skill differences shows regional coherence, suggesting widespread skill improvements south of Cape Hatteras in the dynamic-persistence model, even if not passing a significance test at each specific location. Elsewhere along the coast (i.e., in the Northeast and western Gulf of Mexico), the lead-4 month forecast performance is similar between dynamic persistence and damped persistence. According to the RMSE metric, skill differences between these models at the lead-4 month are minor nearly everywhere (i.e., differences of less than 1 cm; Figure 4b).





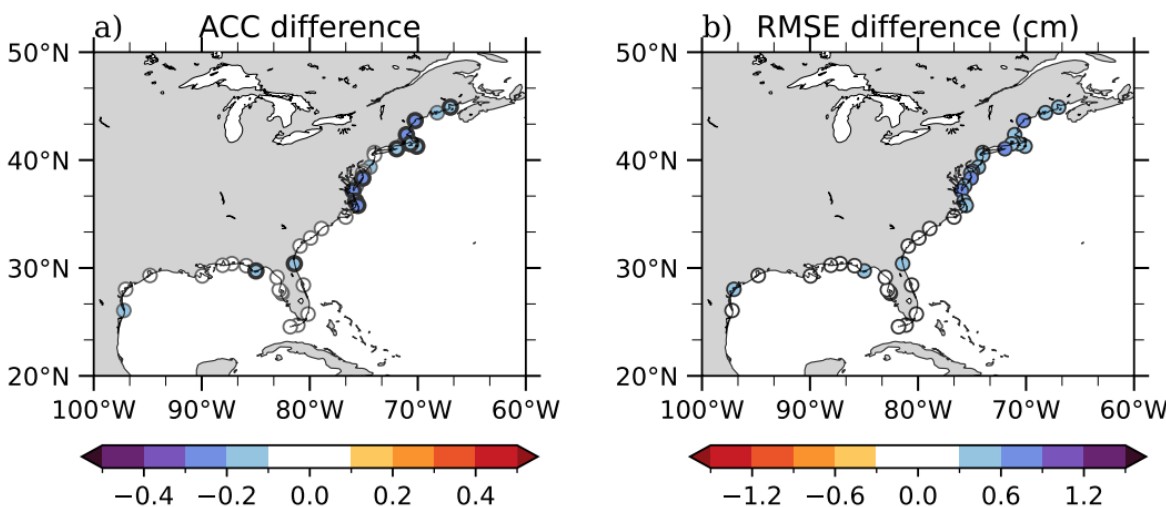

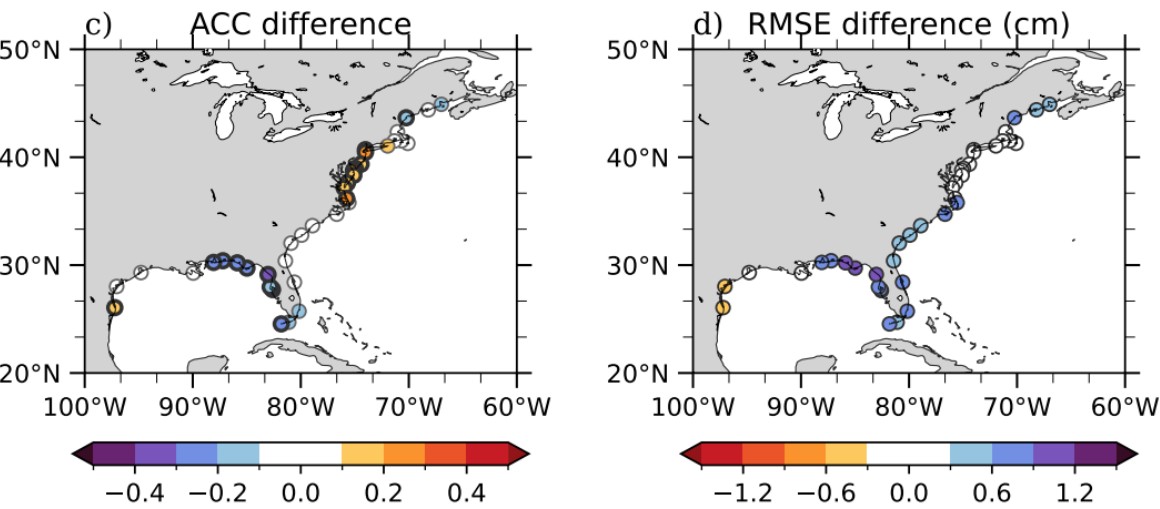

**Figure 3. Differences of retrospective forecast skill (ACC and RMSE) at the lead-1 month for the dynamic-persistence (ECCO; a and b) and climate-forecast (SEAS5; c and d) models compared to damped persistence of monthly sea level anomaly observations by water level gauges (colored circles). Skill of the damped-persistence model is subtracted from that of the other models at each location. Orange shading indicates more skill (higher ACC and lower RMSE) compared to the damped-persistence model. Circles with thick black outlines denote statistically significant ACC differences at the 0.05 significance level.**

210







**Figure 4.** Same as Figure 3, but for the lead-4 month.

215

The climate-forecast model demonstrates utility shortly after its initialization at The Battery and several neighboring locations in the Northeast, with significantly higher ACC than damped persistence at the lead-1 month (Figure 3c, d). However, even at that early lead time, its performance is worse than damped persistence along the northeastern Gulf Coast according to the ACC and RMSE metrics, as well as the Southeast Coast according to only RMSE. At the lead-4 month, there is no evidence that the climate forecast performs better than the other models along the Northeast Coast, nor along the





Gulf and Southeast Coasts (i.e., its ACC and RMSE values are similar or worse than damped persistence at almost all coastal locations considered here; Figure 4c, d).

Changes in forecast skill as a function of lead time are shown in Figure 5 at the example locations for each of the models (the climate forecast ends at the lead-7 month, whereas damped persistence and dynamic persistence extend another 5 months). According to the ACC and RMSE metrics, the climate forecast performs best at only one location (The Battery) but only for the first month (Figure 5a, b). The climate forecast skill decays faster than the other models at all four locations (i.e., the ACC decreases and the RMSE increases). Whereas the skill at the lead-1 month for dynamic persistence is similar to the climate forecast (and damped persistence), the dynamic persistence skill has a slower decay with increasing lead, especially according to the ACC metric (its RMSE worsens at a similar rate as damped persistence). Based on ACC values, dynamic persistence becomes the best-performing model by the lead-5 month at The Battery (Figure 5a), the lead-2 month at Charleston (Figure 5c) as well as Virginia Key (Figure 5e), and the lead-3 month at Grand Isle (Figure 5g), with damped persistence being the similar- or better-performing model at earlier leads. RMSE values are similar for damped persistence and dynamic persistence, with both models generally having smaller errors than the climate forecast (Figure 5b, d, f, h).





**Figure 5. Retrospective forecast skill (ACC and RMSE) as a function of lead time at four example locations (labeled in Figure 1a). Markers (see legend) correspond to the damped-persistence (observed), dynamic-persistence (ECCO), and climate-forecast (SEAS5) models. Also shown are the state estimate (ECCO V4r4) and reanalysis (ORAS5), which are the initial conditions corresponding to the latter two forecasts. Monthly sea level anomalies from each product are compared to observations by water level gauges. Gray shading designates less skill than the damped-persistence model at a particular lead month (lower ACC and higher RMSE).**



Differences in initial conditions seem to be associated with forecast skill variations between the dynamic-persistence and climate-forecast models, at least at the earliest lead time. Model initial conditions are from ECCO for dynamic persistence and ORAS5 for the climate forecast (ACC and RMSE metrics for the respective state estimate and reanalysis are marked at the lead-0 month in Figure 5). At the lead-1 month, there is evidence of the model with higher forecast skill at a particular location also having the better initial condition for that area (e.g., at The Battery where ORAS5 and SEAS5 perform better than the ECCO state estimate and dynamic persistence). We note again though that the SEAS5 advantage at The Battery disappears by the lead-2 month. At the other example locations, the ECCO state estimate performs better than the ORAS5 reanalysis, and the skill for dynamic persistence is likewise better (higher ACC) than SEAS5 at the lead-1 month as well as all longer leads.

To remove the effect of any biased initial conditions from the evaluation of using ocean-dynamic persistence to forecast coastal sea levels, we compared the dynamic-persistence skill to that of damped persistence of the ECCO state estimate instead of observations (Figure 6). The advantage of dynamic persistence compared to damped persistence is much clearer according to this comparison. Previously, we evaluated forecasts compared to damped persistence of observations and noted that the dynamic-persistence model performed worst at the lead-1 month along the Northeast Coast (Figure 3). Comparing dynamic persistence to damped persistence of ECCO itself demonstrates that there is in fact a benefit of including ocean dynamics, even for the Northeast Coast at the lead-1 month (Figure 6a, b). At the lead-4 month, compared to the damped-persistence of ECCO, dynamic persistence has higher ACC values and similar or better RMSE nearly everywhere on the East Coast (Figure 6c, d). The benefit of dynamic persistence is less clear along the Gulf Coast, which possibly points to a limitation of ECCO's coarse resolution in allowing oceanic signals to propagate into the Gulf of Mexico.



## Dynamic persistence (ECCO) - damped persistence (ECCO)

### Lead-1 month

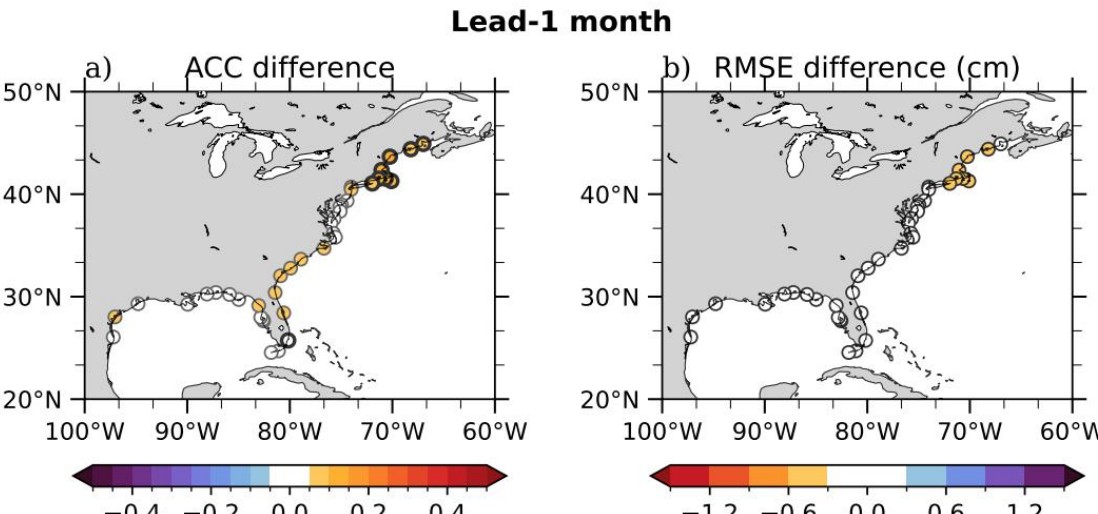

### Lead-4 month

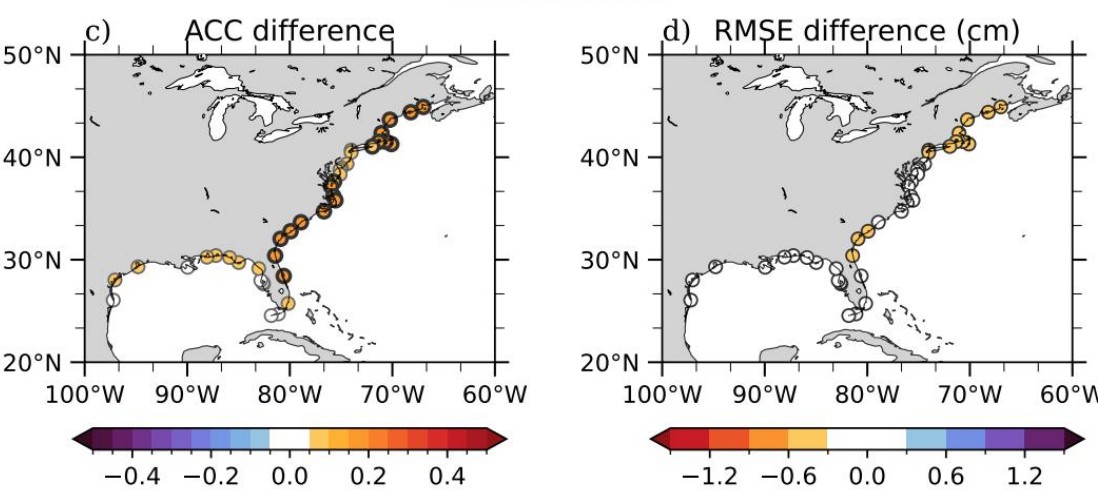

**Figure 6. Difference of skill (ACC and RMSE) for the dynamic persistence (ECCO) and damped persistence (ECCO), with each product being compared to monthly sea level anomalies derived from the ECCO state estimate. Orange shadings indicate more skill (higher ACC and lower RMSE) for dynamic persistence (ECCO) compared to damped persistence (ECCO). Circles with thick black outlines denote statistically significant ACC differences at the 0.05 significance level. Note that smaller ACC differences are more likely to pass the significance test in (a) because the respective ACC values of both models are typically closer to one at the shorter lead time (e.g., at Virginia Key).**

Despite dynamic persistence having the best lead-4 month forecast skill at Charleston, Virginia Key, and Grand Isle, along with most other locations south of Cape Hatteras, we noticed a lack of variability in this model's predicted monthly sea level anomalies. Figures 7 and 8 show the time series of forecasts from each model at the lead-1 and 4 months, respectively,





compared with water level gauge observations. None of the models depict the observed amounts of coastal sea level variability at either lead. For dynamic persistence at the lead-4 month, the SD values are 2.5 cm at The Battery (observed: 5.7 cm), 1.4 cm at Charleston (observed: 6.7 cm), 1.2 cm at Virginia Key (observed: 5.0 cm), and 1.6 cm at Grand Isle
(observed: 5.4 cm). Considering Charleston again for further comparison, variability is only somewhat larger for dynamic persistence at the lead-1 month (2.3 cm); damped persistence similarly suffers from diminished variability, especially by the lead-4 month (0.9 cm; Figure 8b); and, the climate forecast variability is the largest of the models for both the lead-1 and 4 months (4.7 cm and 3.6 cm, respectively). The larger variability of the climate forecast does not translate to better skill at the lead-4 month, because that model deviates the most from observations, as evidenced by it having the lowest ACC and
highest RMSE values for Charleston (0.08 and 7.3 cm; Figure 8b), as well as almost everywhere else on the Gulf and East Coasts (Figure 2c, d). Although dynamic persistence has the best seasonal sea level forecast skill at Charleston and the rest of the Southeast Coast, as well as in parts of the Gulf Coast (Figure 2a, b), its weak variability presents concerns in regards to considering how to utilize forecasts.





**Figure 7. Time series of monthly sea level anomalies according to observations from water level gauges and the lead-1 month retrospective forecasts from the damped-persistence (observed), dynamic-persistence (ECCO), and climate-forecast (SEAS5) models. For each example location (labeled in Figure 1a), the SD (cm), ACC, and RMSE (cm) metrics are indicated for the models and observations (SD only).**




Figure 8. Same as Figure 7, but for the lead-4 month.





## 4 Summary and discussion

This study indicates the potential of using ocean-dynamic persistence to improve seasonal sea level forecasts for the U.S. Gulf and East Coasts, particularly south of Cape Hatteras. By assessing initialized ocean conditions from the ECCO framework that were run forward dynamically under climatological atmospheric conditions, we showed that dynamic

persistence achieves higher forecast skill overall compared to damped persistence. Dynamic persistence also performs similar or better than the SEAS5 climate forecast nearly everywhere that we assessed. Notably, the dynamic-persistence model outperforms the other models at the lead-4 month, with higher ACC values at most locations along the Southeast and Gulf Coasts (Figure 4a). Despite showing minimal improvement in RMSE, dynamic persistence exhibits similar or better skill according to this metric at longer lead times than damped persistence (e.g., at the lead-4 month; Figure 4b), and both of

those models perform better than the much more sophisticated SEAS5 model (Figure 4d).

The better performance of dynamic persistence can be attributed to the ocean's ability to retain the memory of initial conditions, particularly through its high thermal inertia as well as mechanisms such as Rossby waves and horizontal advection, which can influence sea levels over time. This is especially relevant along the Southeast Coast, where delayed

oceanic responses to remote forcings are an opportunity to improve forecast skills (Frederikse et al., 2022). However, because of the climatological forcing of dynamic persistence, initial perturbations in the ocean dissipate over time, which results in much weaker variability of the forecasted sea level anomalies compared to observations. ECCO's coarse resolution (nominally 1°) may also contribute to its weak variability at all leads, and there is emerging evidence that much higher resolution (i.e., finer than 0.25°) is probably necessary to well resolve sea levels along the Gulf and East Coasts (Feng et al.,

2024; Little et al., 2024). Our attempt to scale the dynamic-persistence forecast by its SD did not yield better skill.

Differences between dynamic persistence and damped persistence are less pronounced along the Northeast Coast, which is likely due to the relatively poor initial condition of the former model, as evidenced by the rather low ACC at the lead-0 month for ECCO V4r4 (Figure 5a). Interestingly, when we compare the dynamic-persistence forecast to an experiment using

damped persistence of the ECCO state estimate (instead of observations), much higher skills for dynamic persistence are evident along the Northeast Coast at the lead-1 and 4 months (Figure 6), which points to an opportunity of improving initial conditions as a way toward better forecasts. Yet even at The Battery, dynamic persistence exhibits clearly higher ACC values after the lead-4 month (Figure 5a). It seems that remote signals propagate toward the Northeast Coast and improve the skill of dynamic persistence over time (compared to damped persistence).


While for short lead times the SEAS5 climate forecast has ACC values comparable with the other models (i.e., at the lead-1 month; Figure 1e), and for at least several months longer in some open-ocean regions (Figure 1f), its performance declines rapidly at the coastal locations considered here. The mediocre initial conditions from ORAS5 near the coast (Figure 5; see





also Feng et al., 2024), coupled with the limited predictability of atmospheric variability at seasonal timescales (Lee et al., 2023; Newman et al., 2003), may contribute to SEAS5's lower skill compared to damped persistence and dynamic persistence at the lead-4 month for the Gulf and East Coasts (e.g., Figure 4). The upcoming release of SEAS6, which incorporates data assimilation improvements in coastal areas from the ORAS6 reanalysis (Zuo et al., 2024), may address limitations relating to initial conditions. Plans are also underway to further quantify the role of the ocean observing system in the performance of current subseasonal-to-seasonal forecasting systems (Fujii et al., 2023), which should help to identify aspects of the coastal environment that are most important to assimilate well. For now, this study suggests that dynamic persistence is an alternative and probably better-performing option, compared to the SEAS5 climate model example, for sea level forecasting on some coasts like the regions we assessed. Despite the coarser resolution of the ECCO framework, dynamic persistence seems to capture essential sources of ocean predictability that SEAS5 struggles with, particularly along the Southeast Coast.

Although dynamic persistence shows promise, application in its current form for operational forecasting such as NOAA's monthly high-tide flooding outlook is hindered by its weak variability, which manifests in an inability to predict extreme sea level events. Furthermore, it remains to be tested whether implementing sea level anomalies from the dynamic persistence forecast will improve the high-tide flooding outlook. It is perceivable that concerns about the utility of dynamic persistence in predicting high-tide flooding could be mitigated by improving the model, such as through better initial conditions (i.e., data assimilation improvements) and more realistic physics (i.e., increasing the resolution). Forecast models could also be made more accurate for predicting the coastal sea levels by including the IB effect. Nonetheless, utilizing the dynamic persistence of ECCO or probably other ocean models having realistic initial conditions, would set a higher benchmark than damped persistence and offer a more robust foundation for evaluating the performance of future seasonal forecasting systems. We hope this study encourages further evaluating dynamic persistence as a new baseline for improving seasonal forecasts of sea levels and potentially other aspects of ocean variability.

**Data availability**

Water level gauge observations are available from the NOAA Tides and Currents Archive via the CO-OPS Data Retrieval API (https://api.tidesandcurrents.noaa.gov/api/prod/). Altimetry observations are available from the E.U. Copernicus Marine Service (https://data.marine.copernicus.eu/product/SEALEVEL_GLO_PHY_CLIMATE_L4_MY_008_057/). ORAS5 and SEAS5 are available at https://cds.climate.copernicus.eu/. ECCO State Estimate is available at https://podaac.jpl.nasa.gov/ECCO. ECCO dynamic persistence experiments are available at https://ecco.jpl.nasa.gov/drive/files/Version4/Release4/other/flux-forced/Dyn_Persistence_Forecast.



**Author contribution**

XF, MW, and TL conceived the study. XF and MW wrote the original draft, and all coauthors contributed to the review and editing. XF, OW, MB, and HZ contributed to the data curation. XF contributed to the analysis and production of figures.

**Competing interests**

The authors declare that they have no conflict of interest.

**Acknowledgments**

The authors thank NASA-JPL and ECMWF for making their retrospective forecasts available. This study was primarily supported by the NOAA Climate Program Office's Modelling, Analysis, Predictions, and Projections (MAPP) program through grant NA22OAR4310138. Part of this research was carried out at the Jet Propulsion Laboratory, California Institute of Technology, under a contract with the National Aeronautics and Space Administration (80NM0018D0004). A large-language model (ChatGPT 4o from OpenAI) was used to clarify the writing. This is IPRC publication X and SOEST

contribution Y.

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
