# Peer review of "Indications of improved seasonal sea level forecasts for the United States Gulf and East Coasts using ocean-dynamic persistence"

_EGUsphere, 2025_

## Author Comment (AC1)

In this article the authors explore the use of ocean dynamic persistence for seasonal sea level forecasting on the US east coast. They show that ocean dynamic persistence, based on ECCO-based initialized ocean model forecasts forced with climatological atmospheric forcing show substantial improvements in forecasts of monthly-mean sea level anomalies beyond the first lead month over damped persistence or a full-feature coupled seasonal forecast model.

This is an excellent article that presents a novel approach to ocean forecasting that holds potential for addressing poor seasonal forecasting of sea level on the US east coast. The methods and analysis used are robust, and the results will be of wide interest and applicability. I particularly appreciated the concise presentation without unnecessary complicating detail. I have a series of minor comments that should be considered before publication.

We appreciate reviewer #1's comments, which greatly helped us improve the manuscript. Please see our point-to-point response below in blue and our revisions of the manuscript in red. The line numbers refer to those in the original manuscript.

**General comments:**
* * *
- Can the authors please clarify whether the monthly climatological forcing used in the ECCO dynamical persistence forecasts is based on flux-forcing (e.g. specified wind stresses and heat fluxes) or bulk-forcing (fluxes computed from specified atmospheric state; wind speed, air temperature etc.)? Frederikse et al. use flux-forcing. I presume flux-forcing is used, as bulk forcing based on monthly-averaged atmospheric state will contain significant biases due to non-linearities in the bulk formula (e.g. quadratic wind stress dependence on wind speed). This question is also relevant to whether ocean-sourced dynamical anomalies are damped by the atmospheric forcing (e.g. the statement at line 306). If bulk forcing is used then, for example, ocean SST anomalies will be overly damped toward the climatology, which would then also impact sea level anomalies. On the other hand, with flux forcing there is no damping of ocean anomalies whatsoever, which is also somewhat unrealistic.

The ECCO dynamic persistence forecast is based on a flux-forced ocean model. In this configuration, surface fluxes are calculated externally using atmospheric state variables and prescribed during the model integration. We clarified the model configuration in lines 110-111 as follows:

*"The atmospheric state variables from the ECCO optimization are used to force the ocean model, with surface fluxes of heat, momentum (wind stress), and freshwater being computed externally."*

Under flux forcing, initial sea level anomalies can be damped in two ways. First, open ocean sea level signals forced by atmospheric variability significantly influence coastal sea level anomalies in the study region. When external monthly to interannual atmosphere variability is removed, these forced sea level signals lose their energy source, and the amplitude tends to decay over time and distance due to friction and mixing. Second, the use of climatological atmospheric forcing tends to adjust the ocean towards a steady state, further smoothing out the anomalies. We revised the statement in lines 305-307:

*"However, the dynamic-persistence forecast only captures seasonally-predicable forcings on the ocean, thereby simulating how oceanic initial conditions evolve according to a climatological atmosphere. As a result, sea level variability predicted by dynamic persistence tends to be much weaker compared to what is observed in the real ocean forced by actual atmospheric conditions. The climatological atmospheric forcing used in the dynamic-persistence forecast tends to restore the ocean toward a seasonally steady state, while friction and mixing gradually dissipate existing sea level anomalies, resulting in reduced sea level variability over time (Sérazin et al., 2014)."*

- The authors emphasize that the dynamical persistence forecasts suffer from weak variability, and that if this could be fixed it would improve their skill. To me, this is not an issue with the forecast itself, but a property of the system. The weak variance of the dynamical persistence forecast suggests that the seasonally-predictable signal is a small fraction of the total signal. Unlike an ensemble prediction from a full-feature seasonal forecast model, which includes an ensemble spread as well as an ensemble mean, the ocean-dynamic persistence forecast is designed to only capture the seasonally-predictable signal. Thus this feature could be considered an advantage, not a drawback. I would suggest some reframing of the text to make this point clear (e.g. lines 276-278, 282, 337). This point is also relevant to how the forecasts can be utilized in high-tide flooding outlooks (e.g. lines 336-338). It seems clear that when including predictions for monthly-mean sea level anomalies in these high-tide flooding outlooks, the stochastic component of the monthly-mean sea level still needs to be retained for the purposes of computing, for example, probabilities of threshold exceedance.

We agree that a dynamic persistence forecast captures only the seasonally-predictable (i.e., climatological) forcing on the ocean. Accordingly, the skill of the dynamic-persistence forecast features the minimum skill that a comprehensive coupled forecast system should achieve, compared to damped persistence. The question of why some coupled-forecasting systems perform worse than ocean dynamic persistence would be interesting to investigate in a follow-on study. One possible explanation is that ensemble sizes may be too small to capture the spread of unpredictable atmospheric forcings on the ocean, however, this hypothesis remains to be tested. Regardless, the amount of predictable signal captured by ocean dynamic persistence

depends on the configuration of the forecast system, particularly the initialization and realistic representation of the ocean dynamic processes. We will leave it to the reader to judge whether such characteristics are a strength or weakness of the dynamic-persistence forecast, since the reviewer is correct that this framework excludes the stochastic component of atmospheric forcing, which contributes substantially to monthly sea level variability.

We retained lines 276-278 in the results section and replaced lines 281-283 with the following statement:

*"Potential causes of the weak variability noticed in dynamic-persistence forecasts are discussed in Section 4, along with a mention about how this may concern the usability of such predictions."*

Following the reviewer's suggestion, we also revised lines 305-307 as follows:

*"However, the dynamic-persistence forecast only captures seasonally-predicable forcings on the ocean, thereby simulating how oceanic initial conditions evolve according to a climatological atmosphere. As a result, sea level variability predicted by dynamic persistence tends to be much weaker compared to what is observed in the real ocean forced by actual atmospheric conditions. The climatological atmospheric forcing used in the dynamic-persistence forecast tends to restore the ocean toward a seasonally steady state, while friction and mixing gradually dissipate existing sea level anomalies, resulting in reduced sea level variability over time (Sérazin et al., 2014)."*

**Specific comments:**
* * *
- Lines 33-34; check the grammar. "yet to be" -> "been".

We corrected the sentence as suggested.

- Line 41; "understanding" -> "forecast"?

We corrected the sentence as suggested.

- Line 45; suggest adding "seasonal" or "monthly" in front of sea level anomalies.

Done. The sentence now reads *"…because the statistical persistence of monthly sea level anomalies dampens to near zero by the lead-4 month…"*

- Line 58; "seems reasonable to expect the possibility of achieving" - this is somewhat convoluted. Suggest a rewrite.

We revised the sentence to *"…it may be possible to achieve more skillful seasonal sea level forecasts for the East Coast."*

- Line 60: For non-US readers, can you clarify what is meant by "Southeast coast"? I presume this is mainly referring to the east coast of Florida, not in the Gulf?

We revised line 60 to read: *"it may be possible to achieve more skillful seasonal sea level forecasts for the East Coast"*. Additionally, we added a new figure in Section 2 to clarify the geographic definition of the "Northeast Coast", "Southeast Coast", and the "Gulf Coast".

[Figure]

*Figure 1. Study domain and water level gauge locations. The Northeast Coast extends from Eastport, ME (8410140) to Oregon Inlet Marina, NC (8652587); the Southeast Coast extends from Beaufort, NC (8656483) to Virginia Key, FL (8723214); and the Gulf Coast extends from Vaca Key, FL (8723970) to Port Isabel, TX (8779770). The 7-digit numbers following the location names denote the NOAA station IDs of the water level gauges.*

- Line 61; "likely to continue to" -> again, suggest a rewrite, their influence is not trending.

We revised this to read: "…are likely to complicate efforts to skillfully forecast the sea level variability…"

- Lines 67-74; it may be worth pointing out that Frederikse et al.'s approach is distinct from the approach taken here. They use ECCO adjoint sensitivities, a linear-combinations-of-forcings etc. and thus their approach is not ocean dynamical persistence in the sense examined in this paper. Otherwise, readers may be confused that this paper is just a generalization of that study to the whole East and Gulf coasts.

In lines 67-68, we mentioned that Frederikse's work used the joint sensitivity analysis. To highlight the different approach, inline 77, we add additional explanations: *"Different from Frederikse et al. (2022), where the sea level forecasts rely on a pre-computed sea level sensitivity to atmospheric forcing to represent ocean-dynamical responses,* our investigation utilizes a set

of retrospective forecasts produced with an initialized version of the ECCO model that runs forward for 12 months under climatological atmospheric conditions."

- Lines 86-90: Please mention whether or not detiding is performed on the tide gauge data prior to the computation of monthly means (it shouldn't make much difference).

In lines 90-91, we clarified the processes to remove tide and the generation of the daily time series. *"The hourly time series are averaged to daily data after removing the tide components with the Unified Tidal Analysis and Prediction functions in MATLAB (UTide)."*

- Lines 161-167: It seems to me that this is most likely associated with the low-resolution of the ECCO model (and SEAS5). I would expect that a well-initialized (altimetry assimilating) high-resolution eddy-resolving ocean model making a forecast in dynamical-persistence mode would be able to beat simple damped persistence in the first month in strongly ocean-internal driven dynamics eddying regions. This would be worth mentioning here (i.e. it's specifically *this* ocean dynamical persistence forecast that doesn't perform well here, not all ocean dynamical persistence forecasts).

We agree with the reviewer that a dynamic persistence forecast using an eddy-resolving ocean model with more realistic initialization is presumably to outperform damped persistence in the first month. Lines 161-167 are intended to describe results supported by our current analysis and figures. We discussed the potential influence of model resolution in lines 307-309 of the Discussion section.

- Lines 183: This statement is a bit misleading, since it is difficult to see what is happening with RMSE on the coast given the weak variability and colorbar choice. There is no obvious decay in RMSE with lead time in these plots. Perhaps remove the statement? The same thing goes for the reference to Fig. 2 at line 191.

This paragraph aimed to describe the skill of the damped persistence, as evaluated by both ACC and RMSE. As the reviewer noted, there is no obvious decay in RMSE at lead-4, which is one important feature of the damped persistence and other models. Since this is the first time we discussed the RMSE distribution at the lead-4 and its evolution with lead time, we think it was important to retain this discussion rather than omit it. Including the RMSE evaluation will better explain our conclusion that the decline in forecast skill with lead time mostly refers to ACC, not RMSE.

Following the reviewer's suggestion, we revised line 183 to: *"RMSE is also low in the coastal regions and the changes from lead-1 to lead-4 month are small"* and we removed the statement in line 191.

- Lines 220-223: There is a positive ACC difference in Fig. 4c for a few sites around Delaware, although the differences are not significant. So I'm not sure these statements are completely true.

We revised lines 220-222 to *"At the lead-4 month, there is no wide-spread evidence that the climate forecast performs better than the other models along the Northeast Coast, nor along the Gulf and Southeast Coasts (i.e., its ACC and RMSE values are similar or worse than damped persistence at almost all coastal locations considered here, although the climate forecast has slightly higher ACC at three locations in the Chesapeake Bay area to the north of Cape Hatteras."*

- Line 306-307: As mentioned above, I'm not sure I agree that the dissipation of these anomalies is due to the climatological forcing. Initial ocean perturbations will dissipate over time due to internal ocean dynamics (planetary wave generation and propagation, friction etc.). The reason that the variability is much weaker is just because you have removed a strong source of variability (the weather/non-climatological atmospheric forcing).

Please see our reply to the general comments above. We have revised and clarified the statement as below: *"However, the dynamic-persistence forecast only captures seasonally-predicable forcings on the ocean, thereby simulating how oceanic initial conditions evolve according to a climatological atmosphere. As a result, sea level variability predicted by dynamic persistence tends to be much weaker compared to what is observed in the real ocean forced by actual atmospheric conditions. The climatological atmospheric forcing used in the dynamic-persistence forecast tends to restore the ocean toward a seasonally steady state, while friction and mixing gradually dissipate existing sea level anomalies, resulting in reduced sea level variability over time (Sérazin et al., 2014)."*

- Lines 318-319: Zhu et al. 2024 (cited in the introduction) computed the contribution of remote forcings via wave propagation to sea level anomalies along the east coast, including at seasonal time-scales (e.g. see their Fig. 3). Thus, a more definite statement could likely be made here. Comparison to their results could yield some insights.

We revised line 317-319 to: *"Yet even at The Battery location on the Northeast Coast, dynamic persistence exhibits clearly higher ACC values after the lead-4 month compared to the other models (Figure 7a), which could be due to the propagation of remote signals improving the skill of dynamic persistence over time (at least compared to damped persistence)."*

- Lines 324-326: I wonder whether the relatively low number of ensemble members in the SEAS5 hindcast considered here means that stochastic, non-predictable processes still have a significant impact on the SEAS5 ensemble mean? See general comment #2 above.

Yes, the ensemble size might be too small to capture the atmosphere variability at the seasonal timescale, particularly for the U.S. East Coast wintertime.

---

## Author Comment (AC2)

**Response to Reviewer #2**

We thank Reviewer #2 for the valuable feedback. We have reviewed all comments carefully and incorporated them into our revised manuscript. Our responses are indicated in blue, and modifications to the text in red. The line numbers correspond to those found in the original manuscript.

**General comments:**

This study evaluates the seasonal forecast by statistical and model forecast methods. Among the three forecasts, ocean inertia in the ECCO forecast seems to outperform the other two at lead time of four months. It is interesting but not surprised to see the importance of model initial conditions and ocean inertia. One important unanswered question is why the climate model forecast (SEAS5) including both ocean initial conditions, ocean dynamics and full atmospheric forcing has less predictability than the ECCO forecast without atmospheric forcing. It could be better to have seasonal forecasts with ECCO that include both inertial condition and atmospheric forcing to answer the above question. If there is no such ECCO forecast, it is better to think about some method to illustrate how atmospheric forcing act along with or counteract ocean inertia in the ECCO or SEAS5 model.

We appreciate the reviewer's feedback. Long et al. (2021) demonstrated that climate models can produce skillful seasonal sea level forecasts in the tropical and subtropical open ocean, and SEAS5 is one of the best-performing models. In the coastal regions of the Northwest Atlantic, it's challenging for climate models to generate skillful forecasts, and the potential reasons have been discussed in previous studies (e.g., Frederikse et al., 2022; Long et al., 2025). The skill of climate models depends on the accuracy of initial conditions and representation of atmospheric/oceanic processes and air-sea coupling. With a fully coupled model like SEAS5, it is difficult to determine exactly what causes the low forecast skill. This could be biased from poor initial conditions, unrealistic air-sea coupling, inadequate skill in atmosphere forecasts, or some combination of these factors.

ECCO dynamic persistence forecast is an initialized ocean model running with prescribed climatology atmosphere forcing. It provides an opportunity to evaluate the predictable signal due to ocean dynamics while limiting the direct influence from external atmospheric forcing. We hope that reporting indications of improved sea level forecasting skill using the ocean-dynamic persistence framework will encourage further testing of this method, especially by the modeling centers that produce the operational forecasts. For example, it would be interesting to compare SEAS5 with its ocean-dynamic equivalent, should such an experiment be performed.

The potential reasons for the lack of skill in SEAS5 have been discussed in lines 323-326. Additionally, we revised the manuscript to clarify our motivation further.

In line 40, we added "*For coastal sea levels at such locations, seasonal forecast skill is perhaps limited by inaccurate initializations of the climate models (Feng et al., 2024; Widlansky et al., 2023), simulation biases in the coupled atmosphere-ocean system (Meehl et al., 2021; Roberts et al., 2021) and low predictability of the wind stress anomalies (Newman et al., 2003; Obarein et al., 2023).*"

We revised lines 110-111 to "*The atmospheric state variables from the ECCO optimization are used to force the ocean model, with surface fluxes of heat, momentum (wind stress), and freshwater being computed externally.*"

In line 113, we added "*This flux-forced model configuration allows us to isolate the effect of ocean dynamics on the seasonal sea level forecast, while excluding the potential biased atmosphere forecast and ocean-atmosphere interactions.*"

Regarding the suggestion to explore the role of atmosphere forcing in ECCO dynamic persistence forecast, unfortunately, we currently do not have such experiments (i.e., ECCO forced with a realistic or forecasted atmosphere). Such an investigation is planned, should the necessary experiments be conducted.

**Specific comments:**

One concern is how well ECCO or climate model simulates coastal sea level at the tide gauges along the coast compared to observations, even though the authors shows evaluation at four locations in Fig. 5. For example, both ECCO and SEAS5 have low-resolution so that they may not be able to resolve the coastal sea level variability. It's not a good idea to use one model that cannot simulate the coastal sea level very well.

Model horizontal resolution impacts how well sea level variability is reproduced, especially in the Gulf and East Coast regions. Feng et al. (2024) showed more realistic U.S. East Coast sea level variability associated with high-resolution reanalyses like GLORYS12 and HYCOM. However, eddy-permitting reanalyses like ORAS5 can still reproduce most of the coastal sea level variability (see Fig. 6 in Feng et al., 2024 and the figure below). SEAS5 is the operational forecast system that uses the same models and configuration as ORAS5. We agree with the reviewer that a higher resolution ocean forecast may have better performance, unfortunately, there is no global ocean forecast available at eddy-resolving resolution for the seasonal lead times considered here.

The ECCO dynamic persistence forecast was produced with a similar ocean model configuration as the ECCO state estimate (ECCO V4r4). The ACC of ECCO V4r4 (see the figure below) shows that it has better skill than ORAS5 along the Gulf Coast, comparable skill along the Southeast Coast, and less skill along the Northeast Coast. Despite the weaker performance along the

Northeast Coast, the ECCO dynamic-persistence forecast achieves a higher ACC than SEAS5 after lead-4 month (Fig. 5 in the manuscript). It is perceivable that more sophisticated dynamic persistence frameworks (i.e., with improved initializations and higher model resolutions) will achieve a better skill.

In line 110, we added two sentences and a new figure to demonstrate the skill of ECCO and ORAS5. "*Despite its coarse resolution, ECCO V4r4 can reasonably reproduce coastal sea level variability (Figure 2). Along the Southeast and Gulf Coasts, ECCO V4r4 shows comparable ACC and RMSE to eddy-permitting ocean reanalyses such as ECMWF's Ocean ReAnalysis System 5 (ORAS5; Zuo et al., 2019).*"

[Figure]

*Figure 2. ACC and RMSE of monthly sea level anomalies from the ECCO state estimate and ORAS5 with water level gauge observations. The ORAS5 assessment is modified from Feng et al. (2024).*

Atmospheric forcing cannot be neglected in the sea level forecast. Is there ECCO seasonal forecast that includes atmospheric forcing so that you can compare the roles of atmospheric forcing with ocean inertia? If observed wind forcing make the forecast worse, the bad forecast might be related to some deficiencies of numerical models.

Atmospheric forcing clearly affects the coastal sea level variability (Piecuch et al., 2016; Zhu et al., 2023). However, in an operational seasonal forecast model, the atmospheric forcing is predicted from the coupled climate model. Limited predictability of the atmosphere at seasonal lead times, especially near the U.S. East Coast, likely limits the potential predictability of the ocean. Why SEAS5 does not achieve at least the same skill of ocean dynamic persistence using the ECCO model remains an interesting question, which could be answered perhaps by examining the role of ensemble size, erroneous initial conditions, and/or other modeling deficiencies.

Line 171-172. Comparing Fig. 1a with Fig. 1e seems not supportive for this statement. Pointing out the specific region where the climate model has better performance might be helpful.

The sentence was meant to compare the climate forecast to the dynamic persistence. We revised the sentence to "The climate forecast performs particularly well for a broad area of the subtropical Atlantic Ocean, where its ACC values equal or beat *dynamic persistence*."

Line 256-258. Because sea level in the targeted month is derived from previous months, "damped persistence of ECCO or observation" somehow considered all processes including ocean dynamics but in a statistical way.

A damped-persistence forecast at one location does not account for ocean dynamics, meaning it relies solely on local observations without considering remote information when predicting sea levels. For instance, consider a forecast for some location "X" in the future, a damped-persistence forecast is based exclusively on preceding observations at X, while a dynamic persistence forecast takes into consideration the ocean state in all other locations (both open oceans and other coastal regions), as these non-local signals may be dynamically advanced to location X.

Fig. 6 might also suggest that ECCO data with full forcing (damped ECCO) cannot capture coastal sea level very well. Are there other model forecast results with good performances in simulating coastal sea level?

Assessments of ECCO dynamic persistence before Figure 6 are all compared with observations, despite the dynamic-persistence forecast using the initial condition from the ECCO state estimate. As we stated earlier, the skill of dynamic persistence stems from the initial condition being advanced through ocean dynamical processes. Thus, when we compare dynamic persistence skill with damped persistence of observations, the difference in initial condition and non-local ocean memory both contribute to the skill difference. In Figure 6, the skill assessment is based on verification with the ECCO state estimate instead of observations, meaning that we conduct the forecast assessment in the framework of assuming that the ECCO state estimate is reality. This assessment more clearly isolates the benefit of ocean dynamic persistence, compared to damped persistence. Therefore, Figure 6 demonstrates the advantages of including

the non-local ocean memory (i.e., ocean dynamics) compared to damped persistence (i.e., local ocean memory), because here the two frameworks use the same initial conditions.

We added "*Here, these forecasts are evaluated against the ECCO state estimate.*" to line 254 to make our point clearer.

Line 270-271. The weak variability in this model's predicted monthly sea level anomalies is expected because atmospheric forcing except the climatology is included in the ECCO forecast and ECCO has coarse resolution to resolve coastal sea level. Again, some numerical models have difficulty to simulate coastal sea level variability.

We revised Section 4 and expanded the discussion about explanations and implications of the weak variability of sea level anomalies from the dynamic-persistence forecast.

Lines 306-310: "*However, the dynamic-persistence forecast only captures seasonally-predicable forcings on the ocean, thereby simulating how oceanic initial conditions evolve according to a climatological atmosphere. As a result, sea level variability predicted by dynamic persistence tends to be much weaker compared to what is observed in the real ocean forced by actual atmospheric conditions. The climatological atmospheric forcing used in the dynamic-persistence forecast tends to restore the ocean toward a seasonally steady state, while friction and mixing gradually dissipate existing sea level anomalies, resulting in reduced sea level variability over time (Sérazin et al., 2014). ECCO's coarse resolution (nominally 1°) may also contribute to its weak variability at all leads, and there is emerging evidence that much higher resolution (i.e., finer than 0.25°) is probably necessary to well resolve sea levels along the Gulf and East Coasts (Feng et al., 2024; Little et al., 2024). Our attempt to scale the dynamic-persistence forecast by its SD did not yield better skill according to the RMSE metric.*"

Concerns about "some numerical models have difficulty to simulate coastal sea level variability" are addressed in our response above to Question 1.

Line 301-304. The differences between climate models might also contribute to the forecast differences.

We revised the sentence to clarify the statement:

"The better performance of dynamic persistence compared to damped persistence can be attributed to the ocean's ability to retain the memory of initial conditions, particularly through its high thermal inertia as well as processes such as Rossby waves and horizontal advection, which can influence sea levels over time."

References:

Long, X., Newman, M., Shin, S.-I., Balmeseda, M., Callahan, J., Dusek, G., Jia, L., Kirtman, B., Krasting, J., Lee, C. C., Lee, T., Sweet, W., Wang, O., Wang, Y., and Widlansky, M. J.: Evaluating Current Statistical and Dynamical Forecasting Techniques for Seasonal Coastal Sea Level Prediction, J Clim, 38, 1477–1503, https://doi.org/10.1175/JCLI-D-24-0214.1, 2025.

Meehl, G. A., Richter, J. H., Teng, H., Capotondi, A., Cobb, K., Doblas-Reyes, F., Donat, M. G., England, M. H., Fyfe, J. C., Han, W., Kim, H., Kirtman, B. P., Kushnir, Y., Lovenduski, N. S., Mann, M. E., Merryfield, W. J., Nieves, V., Pegion, K., Rosenbloom, N., Sanchez, S. C., Scaife, A. A., Smith, D., Subramanian, A. C., Sun, L., Thompson, D., Ummenhofer, C. C., and Xie, S.-P.: Initialized Earth System prediction from subseasonal to decadal timescales, Nat Rev Earth Environ, 2, 340–357, https://doi.org/10.1038/s43017-021-00155-x, 2021.

Newman, M., Sardeshmukh, P. D., Winkler, C. R., and Whitaker, J. S.: A Study of Subseasonal Predictability, Mon Weather Rev, 131, 1715–1732, https://doi.org/https://doi.org/10.1175//2558.1, 2003.

Obarein, O. A., Lee, C. C., Smith, E. T., and Sheridan, S. C.: Evaluating Medium-Range Forecast Performance of Regional-Scale Circulation Patterns, Weather Forecast, 38, 1467–1480, https://doi.org/10.1175/WAF-D-22-0149.1, 2023.

Roberts, C. D., Vitart, F., and Balmaseda, M. A.: Hemispheric Impact of North Atlantic SSTs in Subseasonal Forecasts, Geophys Res Lett, 48, e2020GL0911446, https://doi.org/https://doi.org/10.1029/2020GL091446, 2021.

Sérazin, G., Penduff, T., Grégorio, S., Barnier, B., Molines, J.-M., and Terray, L.: Intrinsic Variability of Sea Level from Global Ocean Simulations: Spatiotemporal Scales, J Clim, 28, 4279–4292, https://doi.org/10.1175/jcli-d-14-00554.1, 2014.